# Effects of Cold Temperature and Acclimation on Cold Tolerance and Cannabinoid Profiles of *Cannabis sativa* L. (Hemp)

**Andrei Galic** [1,*], **Heather Grab** [1], **Nicholas Kaczmar** [1], **Kady Maser** [1], **William B. Miller** [1] **and Lawrence B. Smart** [1,2,*]

1   School of Integrative Plant Science, College of Agriculture and Life Sciences, Cornell University, Ithaca, NY 14850, USA; hlc66@cornell.edu (H.G.); nsk37@cornell.edu (N.K.); km755@cornell.edu (K.M.); wbm8@cornell.edu (W.B.M.)
2   School of Integrative Plant Science, College of Agriculture and Life Sciences, Cornell AgriTech, Geneva, NY 14456, USA
*   Correspondence: ag2586@cornell.edu (A.G.); lbs33@cornell.edu (L.B.S.)

**Abstract:** Hemp (*Cannabis sativa*) is a multi-use crop garnering newfound attention from researchers and consumers. While interest has emerged, a lack of substantiated research still exists regarding effects of adverse weather events on physiological health and secondary metabolite production of hemp. The aim of this experiment was to assess cold tolerance of hemp using the cultivars 'FINOLA' and 'AutoCBD'. Effects of cultivar, plant age, cold acclimation, frequency of cold treatments, and intensity of cold treatments were all considered in regard to their influence on physiological stress, biomass, and cannabinoid profile. Few effects of sequential cold treatments were noted, and they were not moderated by cold acclimation, which tended to have negative effects across many responses. This detrimental effect of cold acclimation conditions was further observed in decreased total CBD % and total THC % compared to non-acclimated plants. These findings bear consideration when assessing the unpredictability of a changing climate's effects on the heath and cannabinoid profile of hemp.

**Keywords:** industrial hemp; cold acclimation; chlorophyll fluorescence; freezing damage; abiotic stress; cannabinoids; electrolyte leakage

## 1. Introduction

*Cannabis sativa* L. is an annual, dioecious crop and agricultural commodity produced for its use in textiles, food, and cannabinoid medicine. With its industry projections nearing $6.3 billion by 2025, cannabidiol (CBD) is quickly being integrated into mainstream society [1]. Increasing commercial interest in hemp—*C. sativa* plants containing less than 0.3% Δ9-tetrahydrocannabinol (THC) on a dry-weight basis as per United States federal regulations—has rejuvenated involvement from researchers and industry.

As an emerging crop being rapidly adopted by growers in a wide range of climactic regions, the risk of cultivation in cold temperatures poses several unanswered questions. Freezing damage to crops in the US causes more financial losses than any other weather-related abiotic stress [2]. Toth et al. [3] studied the influence of five abiotic or biotic stresses on cannabinoid accumulation and profile, ultimately concluding these stresses (with the exception of herbicide application) had no significant effect on the production of cannabinoids. However, still very little is documented regarding the timing and intensity of cold temperature abiotic stress effects on the physiological health and total cannabinoid levels of hemp.

Hemp produces a variety of secondary compounds that are most highly concentrated in the capitate stalked trichomes found on the apical inflorescences of female plants [4,5]. This region of the plant is the most abundant producer of CBD, the primary legal cannabinoid gaining commercial interest within the US and abundant in chemotype III hemp

plants [6,7]. Chemotype III plants (high in CBD and low in THC) are selected by hemp cultivators to maintain compliance with federal regulations. While hemp differs from marijuana in its decreased concentration of the intoxicating cannabinoid THC, these plants are classified as the same species [8]. Cannabigerol (CBG) is an additional cannabinoid garnering research attention due to cannabigerolic acid's (CBGA) role as a precursor to cannabidiolic acid (CBDA) and tetrahydrocannabinolic acid (THCA) [9]. Although day-neutral varieties—those that flower independent of day length—exist, hemp is primarily a photoperiodic crop whose flowering structures initiate with the introduction of short days [10]. Accumulation of cannabinoids correlates with time elapsed after terminal flowering [3,11–13]. The timing of harvest of mature hemp flowers, therefore, is largely dependent on the latitude of the cultivation site when grown outdoors.

Plant species develop varying mechanisms of tolerance to abiotic stresses. However, very little published research exists on the cold tolerance of hemp. A University of Vermont trial indicated that hemp plants grown under row cover had higher average temperatures near the soil surface, but CBD concentrations were not different [14]. Another study sought to evaluate cold tolerance of seedlings of nine hemp cultivars based on duration and intensity of cold acclimation periods. Findings indicated that while cold-acclimation-conferred tolerance differed by cultivar, all cultivars experienced cell damage measured via electrolyte leakage during cold acclimation periods of 7 and 14 days and at $4\,^\circ C \pm 1\,^\circ C$ [15]. Plant age and photoperiodism are also factors affecting cold tolerance [16,17]. Therefore, the objectives of this trial were to:

1. Understand the effect of plant age, cultivar, cold acclimation, frequency and intensity of cold, and their interaction on cold tolerance.
2. Evaluate the effects of cold temperatures on post-harvest biomass yield and cannabinoid content.

## 2. Materials and Methods

### 2.1. Experimental Design

Cold stress experiments consisting of four treatment groups were conducted to assess effects on plant health and cannabinoid levels. Day-neutral, chemotype III cultivars 'AutoCBD' and 'FINOLA' were selected for this trial. 'AutoCBD' (Phylos) is a feminized, high-cannabidiol cultivar, and 'FINOLA' is a dioecious, grain cultivar. Male 'FINOLA' plants were culled immediately after staminate flowers were observed. Day-neutral cultivars were selected to accurately stagger planting date treatments. Seeds were planted at 14 day intervals to form three age groups. In doing so, the stage of flower development was unique to each plant age treatment group on the day of cold stress, but all plant age treatment groups were harvested when plants were 75 days old (Figure 1). Half of the plants were exposed to a 10-day cold acclimation treatment to initiate an acclimation response. One week prior to cold stress treatment, a stratified randomization based on height was used to group four biological replicates from each age group into four cold stress groups (n = 16 per group), each receiving a different cold stress treatment. Due to the dioecious nature and low germination of 'FINOLA', there were not enough female 'FINOLA' plants to form a treatment group of non-acclimated older plants, and the acclimated and non-acclimated groups of youngest plants contained 14 'FINOLA' plants each. Treatment groups contained equivalent amounts of plants (n = 4) except for oldest, 'FINOLA' plants (n = 3). As result, 96 'AutoCBD' and 76 'FINOLA' plants split between three age groups were divided into four cold stress treatment groups per acclimation treatment.

Frequency of cold exposure was tested via whole-plant cold stress. Whole-plant cold stress groups consisted of four plants receiving no cold exposure (control), four plants receiving a single cold exposure, four plants receiving two consecutive cold exposures, and four plants receiving three consecutive cold exposures. Each consecutive whole-plant cold stress occurred 24 h following the previous exposure. Plants were exposed to $-0.5\,^\circ C$ for 3 h in darkness and returned to cold-acclimated conditions until harvest. Plants other than those receiving control treatment were exposed to initial whole-plant cold stress on 4 June

2021. A second exposure for two treatment groups followed on 5 June 2021. The remaining treatment group was subject to a final cold exposure that concluded on 6 June 2021. Stress measurements (see Section 2.3 *Measuring Cold Stress Responses*) were taken for each whole-plant cold stress group four days after their respective final exposure. Although control treatments received no cold exposure, their second, comparative measurements occurred on the day following post stress measurements for plants receiving three cold exposures.

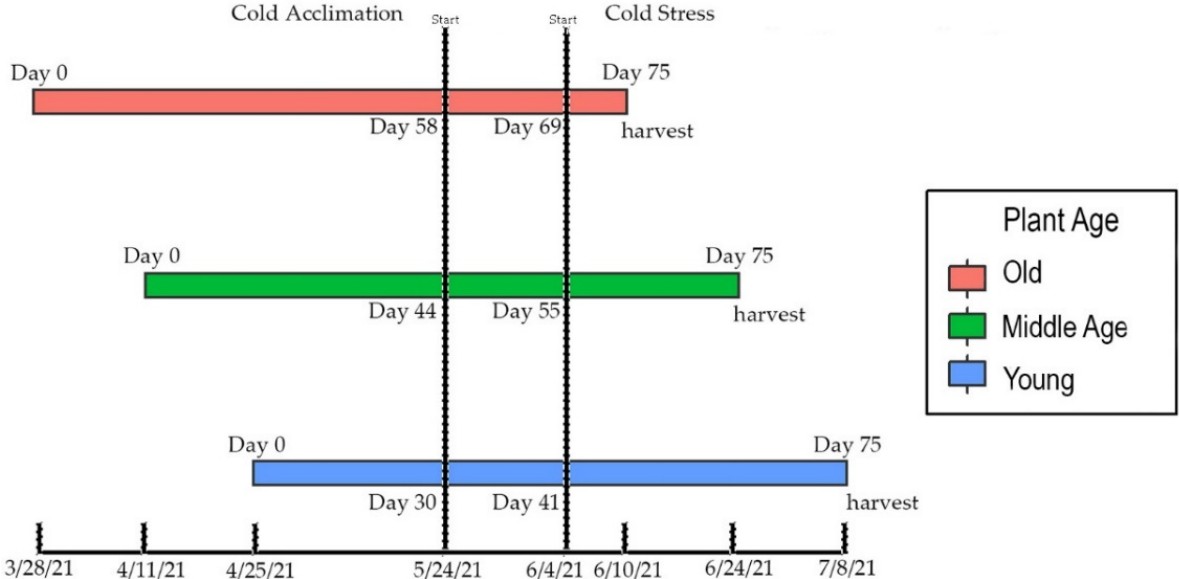

**Figure 1.** Timeline depicting the staggered planting groups in relation to planting, start of their cold acclimation periods and cold stress treatments, and harvest.

Detached-leaf cold stress measures were taken to test the impact of varying intensity of cold temperatures in a freezer. Leaves from the upper half of the plant were detached with petiole intact and placed in plastic bags. Leaves were only sampled from experimental units receiving no whole-plant cold stress for the detached-leaf cold stress experiment. A total of 44 unique leaf samples were placed into the freezing unit for 3-h periods for each temperature treatment. Samples belonging to three distinct treatment groups were exposed to $-2\ ^\circ C$ on 4 June 2021, $-4\ ^\circ C$ on June 5th, 2021, and $-8\ ^\circ C$ on 6 June 2021. Stress measurements were taken immediately after cold exposure. To quantify chlorophyll damage as a result of cold exposure intensity, the same initial Fv/Fm and SPAD values used in quantifying damage resulting from frequency of cold exposure (initial $Fv/Fm_{WP}$ and $SPAD_{WP}$) were used here. However, in this case, $\Delta Fv/Fm_{DL}$ and $\Delta SPAD_{DL}$ values were generated using final Fv/Fm and SPAD values collected immediately after cold exposure (Table 1).

**Table 1.** Formulas used in analysis of chlorophyll fluorescence, SPAD, electrolyte leakage, and cannabinoids.

| Parameter | Formula |
| --- | --- |
| $\Delta Fv/Fm_{WP}$ | (pre-whole-plant-cold-stress Fv/Fm) − (post-whole-plant-cold-stress Fv/Fm) |
| $\Delta Fv/Fm_{DL}$ | (pre-detached-leaf-cold-stress Fv/Fm) − (post-whole-plant-cold-stress Fv/Fm) |
| $\Delta SPAD_{WP}$ | (average pre-whole-plant-cold-stress SPAD) − (average post-whole-plant-cold-stress SPAD) |
| $\Delta SPAD_{DL}$ | (average pre-detached-leaf-cold-stress SPAD) − (average post-whole-plant-cold-stress SPAD) |
| $EL_{WP}$ | (pre-whole-plant-cold-stress EC)/(post-autoclave EC) |
| $EL_{DL}$ | (pre-detached-leaf-cold-stress EC)/(post-autoclave EC) |
| Total CBD % | (CBD %) + (CBDA %*0.877) |
| Total CBG % | (CBG %) + (CBGA %*0.878) |
| Total THC % | (Δ9-THC %) + (THCA %*0.877) |

## 2.2. Growing Environment

All plants were grown in Cornell University greenhouses and growth chambers in Ithaca, NY (Kenneth Post Laboratory). The oldest plants were seeded on 28 March 2021, middle-aged plants on 11 April 2021, and youngest plants on 25 April 2021. Both cultivars were seeded 1 cm deep in 4.5-inch pots containing a commercial all-purpose potting mix (Lambert LM-111). Direct seeding was done to avoid the potential of early flowering caused by transplant shock specific to day-neutral cultivars. Seeds were germinated in a high-humidity propagation greenhouse. Seedlings received overhead misting for 6 s every 15 min. Once seedlings had emerged, plants were fertigated every day with a 21-5-20, 150 ppm fertilizer solution (J. R. Peters, Inc., Allentown, PA, USA). After 14 days in the propagation greenhouse, plants were moved to a different greenhouse where they were grown in ambient light conditions with no supplemental lighting and exposed to daytime temperatures of 22.2 °C and night temperatures of 19.4 °C. All plants received the same fertigation regime as previously described. Cold acclimation period took place in a growth chamber kept at 10 °C with a 14 h light: 10 h dark photoperiod starting 24 May for 10 days. The chamber was equipped with 24 dimmable LED boards (Horticultural Lighting Group) emitting 330 $\mu$mol m$^{-2}$ s$^{-1}$ at bench level. Individual plants were fertigated on an as-needed basis with fertilizer solution kept at ambient growth chamber temperatures. Thrips were targeted with applications of Acephate 97 UP (UPI), Talstar P (FMC), and Safari 20 SG (Valent).

## 2.3. Measuring Cold Stress Responses

Cold stress was quantified via chlorophyll fluorescence, SPAD measurements, and electrolyte leakage. Treatments are referred to as 'cold stress' because cold stress encompasses freezing temperatures, but we cannot be certain that each cold stress caused freezing of all plant tissue. Chlorophyll fluorescence was measured using a LI-6400XT portable photosynthesis system (LI-COR Biosciences, Lincoln, NE, USA). Middle leaflets of detached leaves were dark-adapted for 20 min using dark-adapting clips and subsequently inserted into the 6400-40 Leaf Chamber Fluorometer (LI-COR Biosciences, Lincoln, NE, USA) to measure maximum quantum yield of PSII (Fv/Fm) [18]. Fv/Fm provides valuable insight into photosynthetic capacity of plants as a result of photoinhibition [19,20]. Dark-adapted chlorophyll fluorescence measurements are an especially useful diagnostic in assessing tolerance in relation to freezing damage [21]. Unstressed leaves display Fv/Fm values of ~0.83, and lower values indicate plant stress; therefore, higher values of our parameter measuring change from pre-exposure conditions ($\Delta$Fv/Fm) indicate greater plant stress. Chlorophyll content was estimated using a SPAD 502 Plus Chlorophyll Meter (Konica Minolta, Tokyo, Japan). Photosynthesis can decrease as a result of freeze damage to chlorophyll [22]. Chlorophyll content was approximated as an indirect method to quantify abiotic stress to plants [23–25]. Initial SPAD readings were taken from the middle leaflets of three randomly selected leaves from the upper half of each plant. Chlorophyll parameters were quantified as the change in Fv/Fm ($\Delta$Fv/Fm$_{WP}$) and in SPAD ($\Delta$SPAD$_{WP}$) as measured before and after whole-plant cold stress (Table 1). Initial Fv/Fm and SPAD measurements were taken prior to whole-plant cold stress and also used as initial values for detached-leaf cold stress. Final Fv/Fm and SPAD were taken again 4 days after final whole-plant cold stress and immediately following detached-leaf cold stress.

A modified electrolyte leakage assay was conducted referencing previous protocols [26,27]. Petioles were detached from leaf samples and were placed in 50 mL glass tubes containing 45 mL of deionized water. Tubes were placed on a shaker and agitated at 100 rpm for 16 h. Electrical conductivity (EC) of solution was taken using a HI8733 Multi-range EC Meter (Hanna Instruments, Woonsocket, RI, USA) after agitation. Vessels were autoclaved for 20 min to effectively lyse all cells. After cooling to room temperature, EC readings were taken again. Electrolyte leakage was calculated as the ratio of initial EC measurements divided by the final EC measurements. Plant cells leak electrolytes after damage to membranes [28,29]. Measuring EC before and after cell lysis results in a percentage of leaked electrolytes,

indicating damage due to cold stress. Electrolyte leakage assay was conducted only after the whole-plant and detached-leaf cold stress treatments were completed.

*2.4. Post Harvest Measurements*

Weight and cannabinoid data were collected on all 'AutoCBD' plants. Due to the staggered planting schedule, harvest occurred 75 days after seeding on 6/10, 6/24, and 7/8 for the oldest, middle, and youngest age groups, respectively. Inflorescence samples collected for high-pressure liquid chromatography (HPLC) were collected from the top 10 cm of apical inflorescence and freeze-dried using a Pharma Freeze Dryer (Harvest Right, Salt Lake City, UT, USA). Remaining plant biomass was cut where the stem meets the soil surface and placed in brown paper bags to be dried at ambient greenhouse temperatures. After 10 days of drying in the greenhouse, the dried biomass was weighed. Total dry weight was calculated by adding dry weight of plant biomass with the dry weight of the HPLC sample.

After being stored at $-2$ °C, freeze-dried HPLC samples were granulated by hand to a uniform sample consistency. Individual samples were weighed to 100 mg and mixed with 10 mL of methanol using a VWR Vortexer 2 (Radnor, PA, USA) at room temperature. Samples were diluted 20-fold with methanol and filtered using a Captiva 0.45 μm regenerated cellulose filter. Samples were then analyzed using an Agilent 1220 Infinity II LC system (Santa Clara, CA, USA) using a Poroshell 120 2.7 μm column (3 × 50 mm, Santa Clara, CA, USA). Run conditions included a column temperature of 50 °C beginning with an isocratic 1 mL/min$^{-1}$ ratio of 60:40 methanol + 0.05% formic acid to ultrapure water + 0.1% formic acid for the first minute. This was followed by a 6-min gradient to 77% methanol followed by an additional 90 sec gradient to 95% methanol. UV absorbance was measured at 230 nm. Calibration standards were used for quantification in the range of 1–250 μg mL$^{-1}$ and included THCA, THC, CBDA, CBD, CBGA, and CBG (Agilent). Total percentage of cannabinoids was calculated using formulas in Table 1. Total cannabinoids were measured on a dry-weight basis and analyzed as a percentage. With cannabinoid profile determined by genetic traits, the intent of this study was to investigate chemotype III plants only [3,9]. As such, a single chemotype II 'AutoCBD' plant was not included in cannabinoid analysis to avoid misinterpretation of data.

*2.5. Statistical Analysis*

All statistical analysis was executed using R Studio version 1.4.1717 (R Core Team, Vienna, Austria). A four-way analysis of variance (ANOVA) test was performed to determine if there was an effect of cultivar, acclimation period, plant age, cold frequency or intensity, or an interaction between these factors on the cold tolerance of a plant. Similar models were used to assess cannabinoid profile, though only 'AutoCBD' was measured, so cultivar was not included, resulting in a three-way interaction model. Separate models were fit for each response variable, and distributional assumptions were confirmed by evaluating model diagnostic plots. EL transformation (log) was only required to meet model assumptions prior to analysis. Each full model was simplified by backwards stepwise simplification via the 'step' function in base R. Post hoc Tukey tests were used to evaluate differences among treatment levels via the 'emmeans' function in the package emmeans [30]. Histograms (Supplementary File Figure S1) of residuals for response variables validate model assumptions and justify use of ANOVA.

## 3. Results

### 3.1. Frequency of Cold Exposure in Whole Plants

Cold tolerance, as measured by $\Delta Fv/Fm_{WP}$, increased with cold acclimation (F = 101.466, $p < 0.0001$), decreased with plant age (F = 53.490, $p < 0.0001$), and differed between cultivars (F = 6.365, $p < 0.05$). The oldest plants experienced greater average $\Delta Fv/Fm_{WP}$ than middle-age and youngest plants for both cold-acclimated and non-acclimated plants (Figure 2). Significant two-way interactions existed, but none in-

volved cold treatment (Table 1). Although whole-plant cold stress (F = 2.409, *p* = 0.0695) was the only treatment without a significant main effect, its three-way interaction with both plant age and exposure to cold acclimation was significant (t = 4.228, *p* < 0.001). Control plants for the oldest non-acclimated 'AutoCBD' plants had greater $\Delta Fv/Fm_{WP}$ than one cold exposure (t = 4.254, *p* < 0.001), two cold exposures (t = 3.338, *p* < 0.05), or three cold exposures (t = 4.658, *p* < 0.0001). However, these effects were not observed in other cultivar and cold acclimation treatment interactions. $\Delta SPAD_{WP}$ was influenced by cultivar (F = 13.052, *p* < 0.001) and plant age (F = 9.102, *p* < 0.001) (Table 1). The largest range of values within this interaction of treatments occurred within non-acclimated 'AutoCBD' plants (Supplementary File Table S1). No effects of any cold treatment were observed.

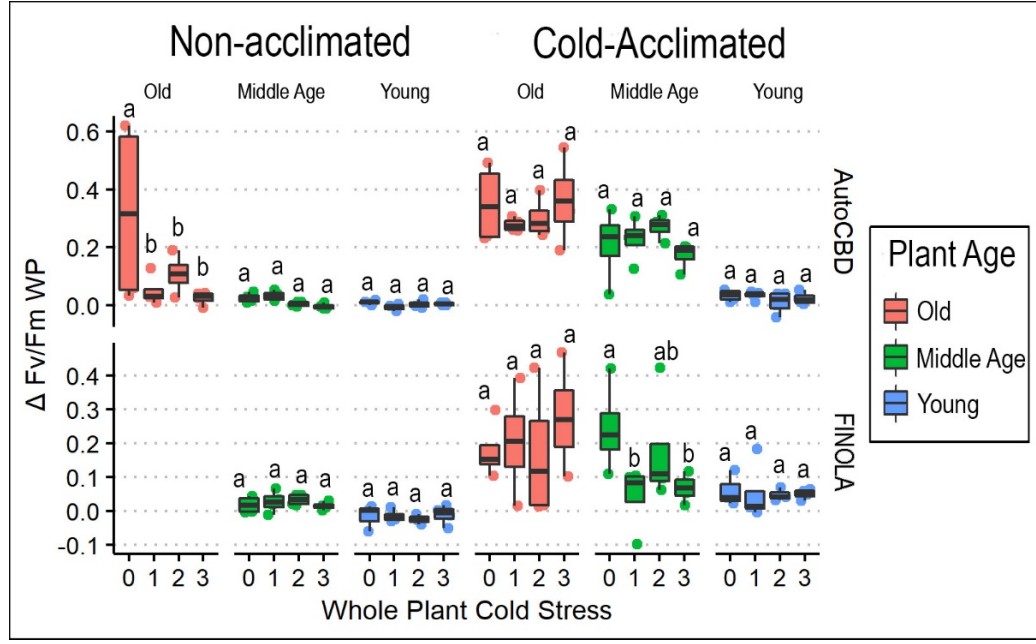

**Figure 2.** $\Delta Fv/Fm_{WP}$ box plots comparing whole-plant cold stress and plant age for cold-acclimated and non-acclimated 'AutoCBD' and 'FINOLA' plants. Data represent the difference between Fv/Fm taken before and after whole-plant cold stress. Non-acclimated plants received no cold acclimation period, and cold-acclimated plants were subject to 10 days at 10 °C. Cold stress consisted of no exposure, **0** (control); a single 3 h −0.5 °C exposure, **1**; two consecutive 3 h −0.5 °C exposures separated by 24 h, **2**; and three 3 h −0.5 °C exposures separated by 24 h, **3**. Plant age treatments include oldest plants, middle-age plants, and youngest plants (68, 54, and 40 days old on the day of whole-plant cold stress, respectively). Y axis scale differs between acclimation treatments to accurately reflect data ranges. Lowercase letters indicate significant differences (*p* < 0.05) between treatments (Tukey's test).

In whole plants exposed to cold, cold tolerance measured by electrolyte leakage ($EL_{WP}$) declined with plant age (F = 29.3715, *p* < 0.0001) and varied with whole-plant cold stress treatment (F = 27.452, *p* < 0.0001) (Table 2). Cold tolerance was also affected through interactions between whole-plant cold stress and cold acclimation (F = 4.768, *p* < 0.001). Whole-plant cold stress had multiple three-way interactions between plant age and cultivar (F = 2.330. *p* < 0.05), and plant age and cold acclimation (F = 2.475, *p* < 0.05). Cold-acclimated plants consistently displayed higher electrolyte leakage measures between control whole-plant cold stress and one cold exposure in the oldest (t = 3.383, *p* < 0.05) and middle-age groups (t = 5.712, *p* < 0.0001) but not in the youngest. Additionally, non-acclimated and cold-acclimated plants both had lower $EL_{WP}$ compared to the groups with one and two cold exposures. In both treatment groups for cold acclimation, the oldest plants of both cultivars experienced a higher percentage of $EL_{WP}$ than middle-age or younger plants (Figure 3). Furthermore, whole-plant cold stress control treatment consistently displayed

higher electrolyte leakage than plants receiving one and two exposure(s) of whole-plant cold stress, and often the same was true in plants receiving three exposures of whole-plant cold stress.

**Table 2.** Summary of effects of cultivar, cold acclimation, plant age, whole-plant cold stress, and their interactions and mean values for 'AutoCBD' and 'FINOLA'. Cold tolerance was quantified with $\Delta Fv/Fm_{WP}$ (difference between Fv/Fm taken prior to and after whole-plant cold stress), $\Delta SPAD_{WP}$ (difference in mean SPAD values taken prior to whole-plant cold stress and after whole-plant cold stress), and EL (ratio between EC of leaf sample after whole-plant cold stress and EC of leaf sample after cell lysis).

| Whole Plant Cold Tolerance | $\Delta Fv/Fm_{WP}$ | $\Delta SPAD_{WP}$ | $EL_{WP}$ |
|---|---|---|---|
| AutoCBD | 0.12 | 34.43 | 0.17 |
| FINOLA | 0.08 | 32.75 | 0.17 |
| Cultivar | * | *** | n.s. |
| Cold Acclimation | *** | n.s. | n.s. |
| Plant Age | *** | *** | *** |
| Whole-Plant Cold Stress | n.s. | n.s. | *** |
| Cultivar × Cold Acclimation | * | n.s. | n.s. |
| Cultivar × Plant Age | n.s. | *** | * |
| Cold Acclimation × Plant Age | *** | n.s. | *** |
| Cultivar × Whole-Plant Cold Stress | n.s. | n.s. | n.s. |
| Cold Acclimation × Whole Plant | n.s. | n.s. | ** |
| Cold Stress Plant Age × Whole-Plant Cold Stress | n.s. | n.s. | n.s. |
| Cultivar × Cold Acclimation × Plant Age | * | n.s. | * |
| Cold Acclimation × Plant Age × Whole-Plant Cold Stress | *** | n.s. | * |
| Cultivar × Plant Age × Whole-Plant Cold Stress | n.s. | n.s. | |

n.s. = not significant; * $p < 0.05$; ** $p < 0.01$; *** $p < 0.001$.

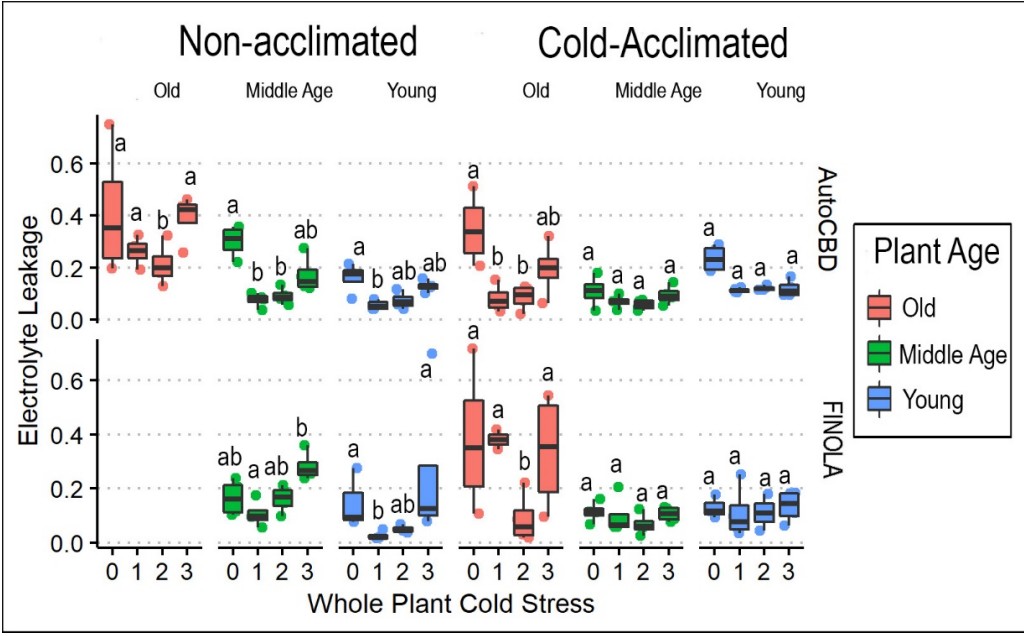

**Figure 3.** $EL_{WP}$ box plots comparing whole-plant cold stress and plant age for cold-acclimated and non-acclimated 'AutoCBD' and 'FINOLA' plants. Electrolyte Leakage represents the ratio between electrical conductivity (EC) of leaf sample after whole-plant cold stress and EC of leaf sample after cell lysis. Non-acclimated plants received no cold acclimation period, and acclimated plants were subject to 10 days at 10 °C. Cold stress consisted of no exposure, **0** (control); a single 3 h −0.5 °C exposure, **1**; two consecutive 3 h −0.5 °C exposures separated by 24 h, **2**; and three 3 h −0.5 °C exposures separated by 24 h, **3**. Plant age treatments include oldest plants, middle-age plants, and youngest plants (68, 54, and 40 days old on the day of whole-plant cold stress, respectively). Lowercase letters indicate significant differences ($p < 0.05$) between treatments (Tukey's test).

### 3.2. Intensity of Cold Exposure in Detached Leaves

Cultivar (F = 8.542, $p < 0.05$), cold acclimation (F = 14.921, $p < 0.001$), and detached-leaf cold stress (F = 31.424, $p < 0.0001$) all directly impacted cold tolerance (Table 3) and interacted with cold stress to impact $\Delta Fv/Fm_{DL}$ (F = 6.583, $p < 0.05$). With the exception of non-acclimated 'FINOLA' plants, $-8\,°C$ showed higher values compared to $-2°\,C$ and $-4\,°C$ in plants of non-acclimated 'AutoCBD' (t = $-4.156$, $p < 0.001$, t = $-4.087$, $p < 0.001$), cold-acclimated 'AutoCBD' (t = $-5.022$, $p < 0.001$, t = $-4.618$, $p < 0.0001$), and cold-acclimated 'FINOLA' (t = $-3.633$, $p < 0.001$, t = $-3.451$, $p < 0.05$). On average, $-8\,°C$ had the highest $\Delta Fv/Fm_{DL}$ values (Figure 4).

**Table 3.** Summary of effects of cultivar, cold acclimation, plant age, detached-leaf cold stress, and their interactions and mean values for plant health values of 'AutoCBD' and 'FINOLA'. Cold tolerance was quantified with $\Delta Fv/Fm_{DL}$ (difference between Fv/Fm taken prior to whole-plant cold stress and after detached-leaf cold stress), $\Delta SPAD_{DL}$ (difference in mean SPAD values taken prior to whole-plant cold stress and after detached-leaf cold stress), and EL (ratio between EC of leaf sample after detached-leaf cold stress and EC of leaf sample after cell lysis).

| Detached Leaf Cold Tolerance | $\Delta Fv/Fm_{DL}$ | $\Delta SPAD_{DL}$ | $EL_{DL}$ |
|---|---|---|---|
| AutoCBD | 0.16 | 0.88 | 0.23 |
| FINOLA | 0.07 | 3.96 | 0.13 |
| Cultivar | ** | *** | *** |
| Cold Acclimation | *** | *** | n.s. |
| Plant Age | n.s. | n.s. | * |
| Detached-Leaf Cold Stress | *** | *** | *** |
| Cultivar × Plant Age | ** | *** | n.s. |
| Cold Acclimation × Plant Age | * | ** | ** |
| Cultivar × Detached-Leaf Cold Stress | * | * | * |
| Cold Acclimation × Detached-Leaf Cold Stress | n.s. | n.s. | ** |
| Cultivar × Cold Acclimation × Plant Age | n.s. | * | n.s. |
| Cultivar × Plant Age × Detached-Leaf Cold Stress | n.s. | *** | n.s. |
| Cultivar × Cold Acclimation × Plant Age × Detached-Leaf Cold Stress | * | n.s. | n.s. |

n.s. = not significant; * $p < 0.05$; ** $p < 0.01$; *** $p < 0.001$.

Detached-leaf SPAD readings were affected by cultivar (F = 34.410, $p < 0.0001$), cold acclimation (F = 16.411, $p < 0.001$), and detached-leaf cold stress (F = 10.640, $p < 0.0001$). Two interactions, cultivar × plant age × detached-leaf cold stress (F = 5.545, $p < 0.001$) and cultivar × detached-leaf cold stress (F = 4.090, $p < 0.05$), occurred involving the effect of cold temperature intensity on cold tolerance. Letters indicate differences amongst consecutive cold exposures between cultivars, plant age, and acclimation (Figure 5).

Electrolyte leakage as a result of cold intensity treatments ($EL_{DL}$) differed between cultivars (F = 17.108, $p < 0.0001$), plant age groups (F = 3.870, $p < 0.05$), and detached-leaf cold stress exposures (F = 12.316, $p < 0.0001$) (Table 3). Interactions among treatment groups were also observed in cultivar × detached-leaf cold stress (F = 4.086, $p < 0.05$), and cold acclimation × detached-leaf cold stress (F = 6.962, $p < 0.01$). Plants receiving $-8°C$ treatment most commonly had highest average EL values in both acclimated 'AutoCBD' (t = $-4.921$, $p < 0.0001$) and 'FINOLA' (t = $-2.657$, $p < 0.05$) across plant age and acclimation treatments (Supplementary File Table S2).

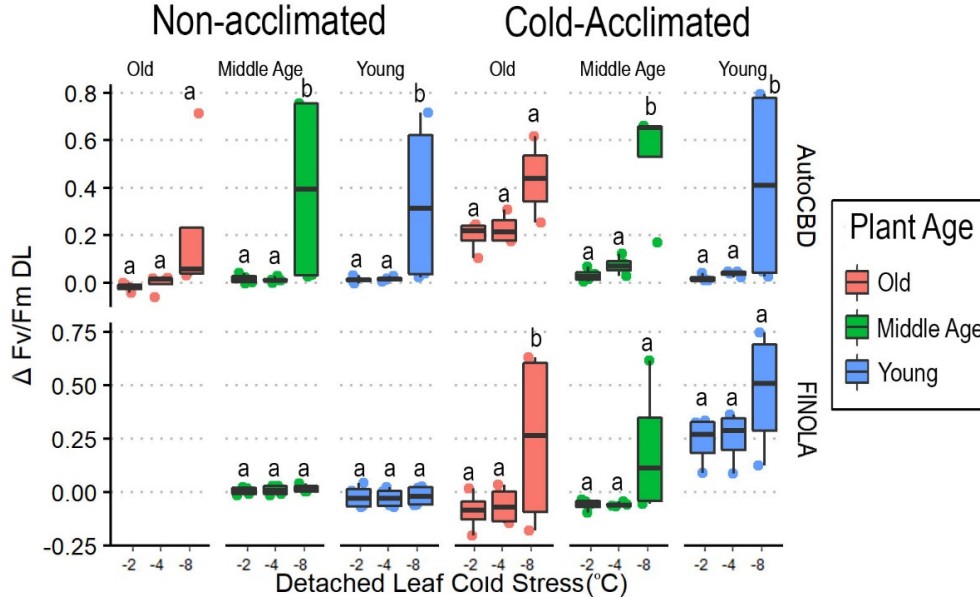

**Figure 4.** $\Delta$Fv/Fm$_{DL}$ box plots comparing detached-leaf cold stress and plant age in cold-acclimated and non-acclimated 'AutoCBD' and 'FINOLA' plants. Data represent the difference between Fv/Fm taken before whole-plant cold stress and after detached-leaf cold stress. Non-acclimated plants received no cold acclimation period, and cold-acclimated plants were subject to 10 days at 10 °C. Detached-leaf cold stress consisted of a single 3 h −2 °C exposure, a single 3 h −4 °C exposure, and a single 3 h −8 °C exposure. Plant age treatments include oldest plants, middle-age plants, and youngest plants (68, 54, and 40 days old on the day of whole-plant cold stress, respectively). Y axis scale differs between acclimation treatments to accurately reflect data. Lowercase letters indicate significant differences ($p < 0.05$) between treatments (Tukey's test).

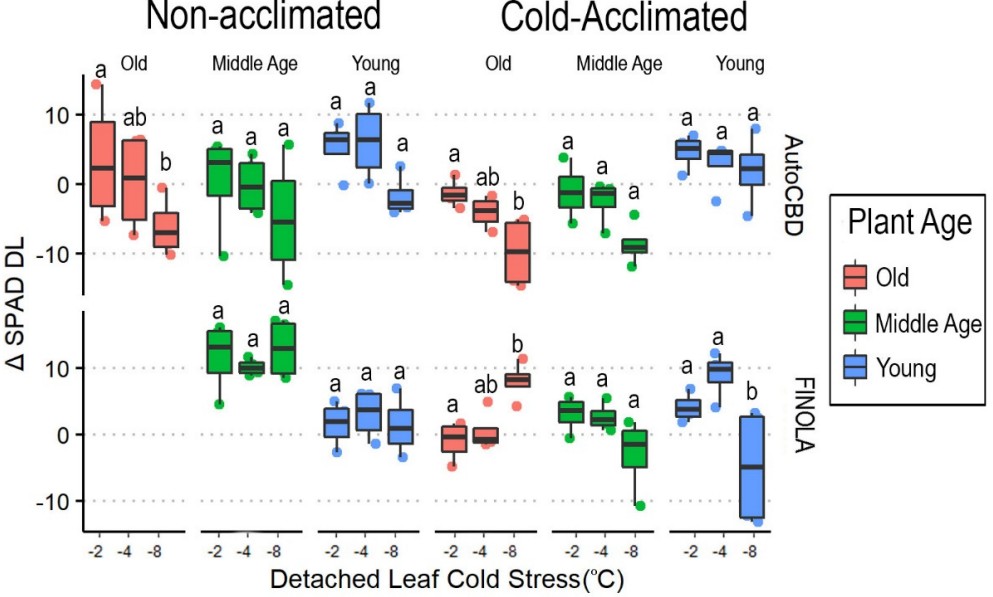

**Figure 5.** $\Delta$SPAD$_{DL}$ box plots comparing detached-leaf cold stress and plant age. Data represent the difference in mean SPAD taken before whole-plant cold stress and after detached-leaf cold stress. Detached-leaf cold stress consisted of a single 3 h −2 °C exposure, a single 3 h −4 °C exposure, and a single 3 h −8 °C exposure. Plant age treatments include oldest plants, middle-age plants, and youngest plants (68, 54, and 40 days old on the day of whole-plant cold stress, respectively). Lowercase letters indicate significant differences ($p < 0.05$) between treatments (Tukey's test).

### 3.3. Cannabinoids and Weight

Postharvest data were collected from only 'AutoCBD' plants (Supplementary File Table S3). Total CBD % decreased with cold acclimation (F = 119.173, $p < 0.0001$). Although no main effect of cold treatment was observed, there was an interaction between cold acclimation, plant age, and cold treatment (F = 3.801, $p < 0.01$). Comparing the total CBD % showed that plants receiving no acclimation treatment had approximately twice the mean total CBD than the acclimation treatment group (Table 4, Figure 6). Similar effects of cold acclimation (F = 25.633, $p < 0.0001$) and plant age (F = 4.603, $p < 0.05$) were observed on total CBG after cold exposure, in addition to an effect of cold stress treatment (F = 2.912, $p < 0.05$). CBG was the cannabinoid that showed the fewest responses to treatments (Figure 7).

In contrast to CBG, the mean total percentages of CBD and THC were more than twice as high in non-acclimated plants compared to those receiving cold acclimation (Table 4). Out of the 94 plants sampled, 26 plants (28%) exceeded regulatory limits for THC (total THC > 0.3%). Cold stress reduced both CBD and THC content but only in the non-acclimated plants exposed to two cold stresses (Figures 6 and 8). Furthermore, the mean ratio between total CBD and total THC was greater in cold-acclimated (27.55) compared to non-acclimated (21.50) plants (Table 4). Weight was reduced by cold acclimation (F = 122.661, $p < 0.0001$) as exemplified by comparison of mean weights in non-acclimated (9.60 g) and cold-acclimated (6.54 g) plants.

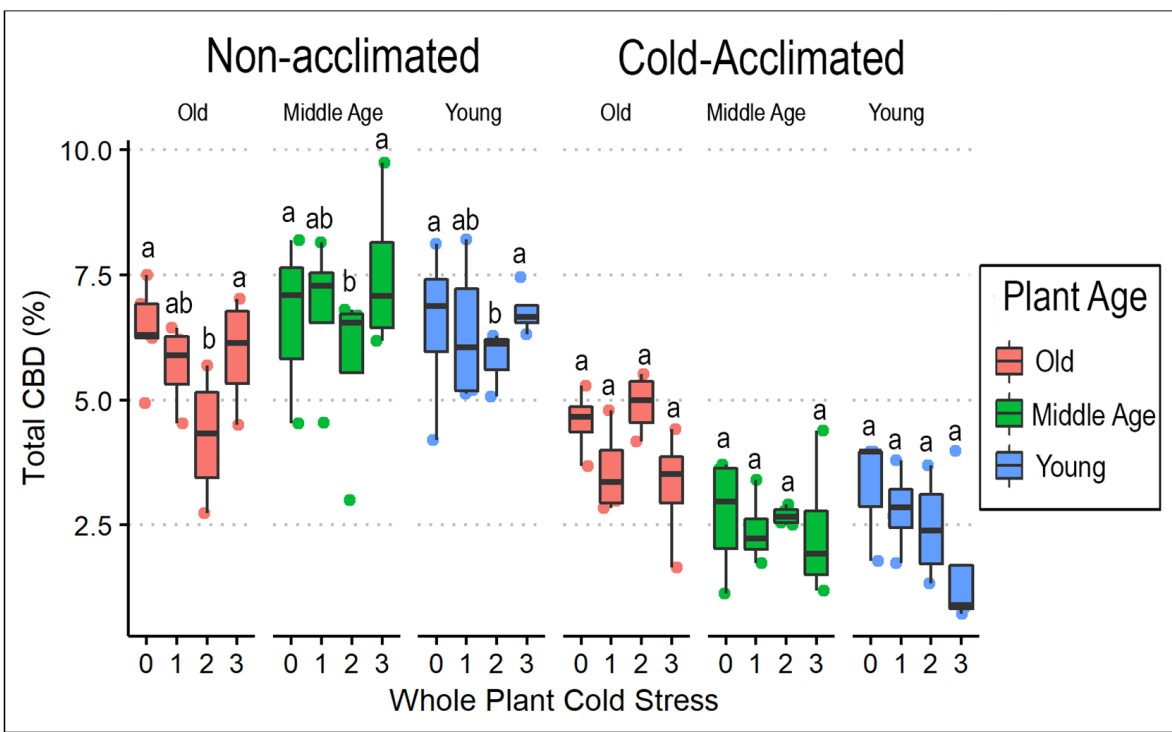

**Figure 6.** Total CBD (%) box plot comparing whole-plant cold stress to acclimation treatment and plant age in 'AutoCBD'. Data represent the percentage of CBD's neutral form added to the product of the percentage of CBD acid form multiplied by the cannabinoid's molecular weight (0.877). Plants receiving cold acclimation treatment were subject to 10 days at 10 °C. Whole-plant cold stress consisted of a control treatment receiving no cold exposure, **0** (control); a single 3 h −0.5 °C exposure, **1**; two consecutive 3 h −0.5 °C exposures separated by 24 h, **2**; and three 3 h −0.5 °C exposures separated by 24 h, **3**. Plant age treatments include oldest plants, middle-age plants, and youngest plants (68, 54, and 40 days old on the day of whole-plant cold stress, respectively). Lowercase letters indicate significant differences ($p < 0.05$) between treatments (Tukey's test).

**Table 4.** Summary of effects of cultivar, cold acclimation, plant age, whole-plant cold stress, and their interactions and mean values for post-harvest properties of 'AutoCBD'.

| Mean | Total CBD (%) | Total CBG (%) | Total THC (%) | Total CBD:Total THC | Weight (g) |
|---|---|---|---|---|---|
| No Acclimation | 6.20 | 0.20 | 0.29 | 21.50 | 9.60 |
| Cold Acclimation | 3.04 | 0.13 | 0.12 | 27.55 | 6.54 |
| Old | 4.88 | 0.19 | 0.21 | 23.54 | 7.98 |
| Middle Age | 4.53 | 0.16 | 0.20 | 23.95 | 7.96 |
| Young | 4.44 | 0.14 | 0.20 | 26.21 | 8.06 |
| Control | 5.07 | 0.20 | 0.22 | 23.83 | 7.69 |
| 1 Cold Exposure | 4.61 | 0.16 | 0.20 | 24.69 | 8.12 |
| 2 Cold Exposures | 4.25 | 0.14 | 0.18 | 24.34 | 7.88 |
| 3 Cold Exposures | 4.58 | 0.14 | 0.21 | 25.20 | 8.30 |
| Cold Acclimation | *** | *** | *** | *** | *** |
| Plant Age | n.s. | * | n.s. | *** | n.s. |
| Cold Stress | n.s. | * | n.s. | n.s. | n.s. |
| Cold Acclimation × Plant Age | *** | ** | *** | *** | * |
| Cold Acclimation × Cold Stress | ** | n.s. | *** | * | n.s. |

n.s. = not significant; $* \ p < 0.05$; $** \ p < 0.01$; $*** \ p < 0.001$.

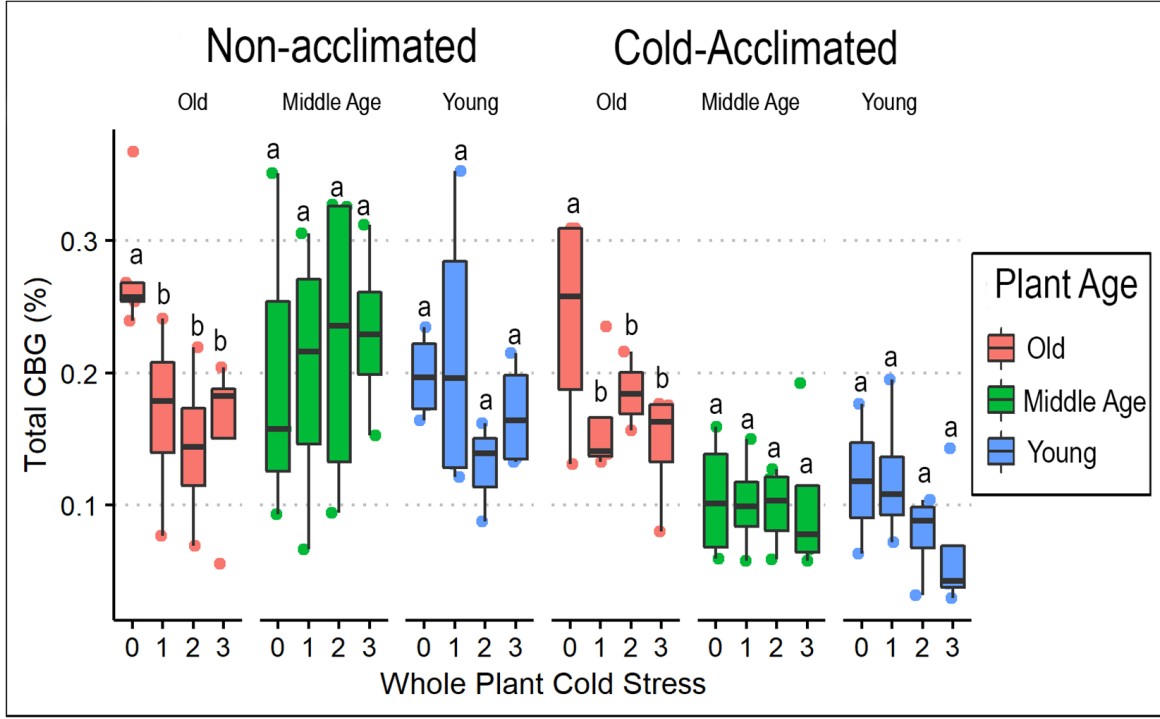

**Figure 7.** Total CBG (%) box plots comparing whole-plant cold stress to acclimation treatment and plant age in 'AutoCBD'. Data represent the percentage of CBG added to the product of the percentage of CBGA multiplied by the cannabinoid's molecular weight ratio (0.878 and 0.877). Plants receiving cold acclimation treatment were subject to 10 days at 10 °C. Whole-plant cold stress consisted of a control treatment receiving no cold exposure, **0** (control); a single 3 h −0.5 °C exposure, **1**; two consecutive 3 h −0.5 °C exposures separated by 24 h, **2**; and three 3 h −0.5 °C exposures separated by 24 h, **3**. Plant age treatments include oldest plants, middle-age plants, and youngest plants (68, 54, and 40 days old on the day of whole-plant cold stress, respectively). Letters indicate differences amongst consecutive cold exposures between cultivars, plant age, and acclimation. Lowercase letters indicate significant differences ($p < 0.05$) between treatments (Tukey's test).

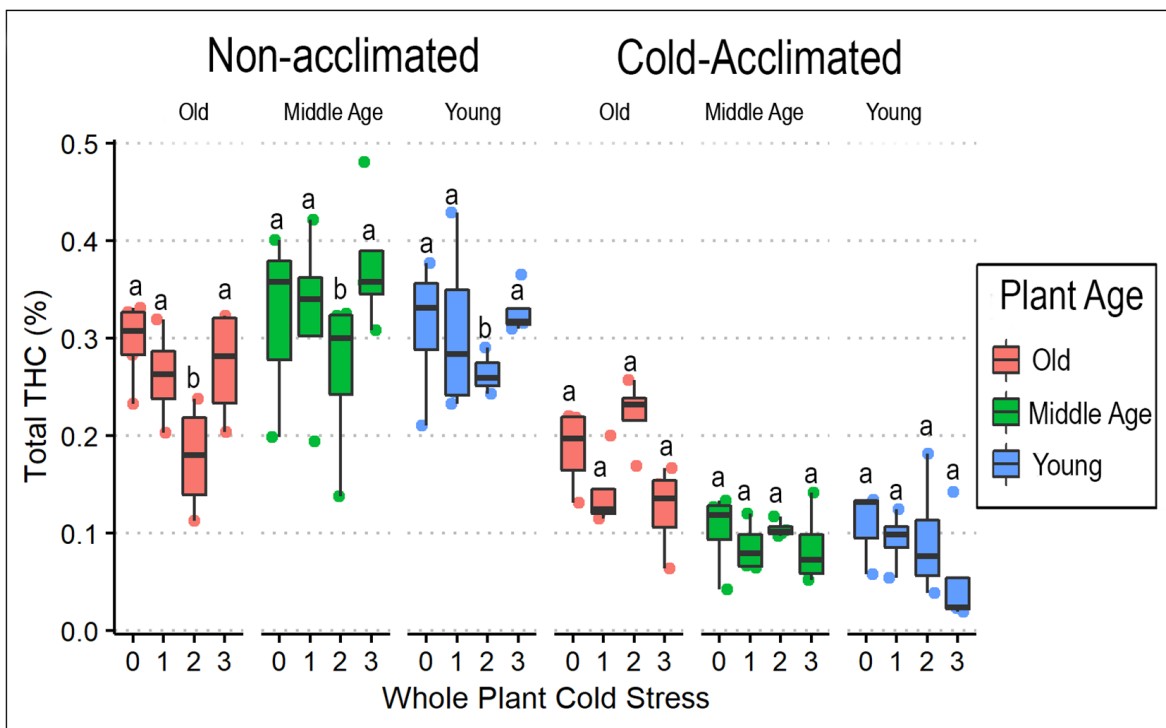

**Figure 8.** Total THC (%) box plots comparing whole-plant cold stress to acclimation treatment and plant age in 'AutoCBD'. Data represent the percentage of THC added to the product of the percentage of THCA multiplied by the cannabinoid's molecular weight ratio (0.878). Plants receiving cold acclimation treatment were subject to 10 days at 10 °C. Whole-plant cold stress consisted of a control treatment receiving no cold exposure, **0** (control); a single 3 h −0.5 °C exposure, **1**; two consecutive 3 h −0.5 °C exposures separated by 24 h, **2**; and three 3 h −0.5 °C exposures separated by 24 h, **3**. Plant age treatments include oldest plants, middle-age plants, and youngest plants (68, 54, and 40 days old on the day of whole-plant cold stress, respectively). Letters indicate differences amongst consecutive cold exposures between cultivars, plant age, and acclimation. Lowercase letters indicate significant differences ($p < 0.05$) between treatments (Tukey's test).

## 4. Discussion

Understanding the effects of cold temperatures on plant health and production of secondary metabolites in a crop grown in many regions is essential in combating uncertainties posed by a changing climate; however, few studies have addressed this topic. Here, we find that non-acclimated plants and young plants across both cultivars exhibited the least amount of physiological stress in response to consecutive cold stress treatments, though consecutive exposures to cold temperatures did not consistently result in greater amounts of photoinhibition (determined via chlorophyll fluorescence and electrolyte leakage) in any treatment group. In combination with these findings, we also observed that 'AutoCBD' plants subject to the cold acclimation period produced significantly less CBD, CBG, and THC and yielded less biomass, suggesting that cold acclimation acted as a plant stress as opposed to a protective, hardening mechanism. Furthermore, detached leaves exposed to −8 °C received a magnitude of damage not observed at either −4 °C or −2 °C across any method of plant stress quantification. Together, these findings suggest that hemp may be quite tolerant to short periods of frost prior to harvest, though prolonged cold weather may reduce overall yields.

### 4.1. Cold Temperature Effects on Plant Health

Based on whole-plant values of $\Delta Fv/Fm_{WP}$ and $EL_{WP}$, this experiment suggests that consecutive cold exposures did not elicit increased plant stress in a manner that was consistent in oldest and cold-acclimated plants. Previous research has indicated that

cold tolerance increases as plants age [16,31,32]. However, the data observed in this trial indicated an opposite effect: mean $\Delta Fv/Fm_{WP}$, $\Delta SPAD_{WP}$, and $EL_{WP}$ data were highest for the oldest plants of both cultivars. The conflicting results may be partly attributed to an observed higher incidence of pest pressure present during the early stages of growth when the oldest planting group was the only age group present in the growing environment (no data shown). If this is the case, damage from cold exposures to hemp may exacerbate stressors such as pests or phytotoxicity of pesticide application. Future research should explore the potential for multiple stressors to interact in driving plant health. Furthermore, control plants receiving no cold exposures often displayed values of plant stress greater than plants receiving one, two, and even three cold exposures. This discrepancy in results may be attributed to the timing of the second comparative measurements taken in control plants and used in calculating $\Delta Fv/Fm_{WP}$, $\Delta SPAD_{WP}$, and $EL_{WP}$. These control measurements were taken three days after post-stress measurements in plants receiving one cold exposure, two days after plants receiving two cold exposures, and one day after plants receiving three cold exposures. As such, an additional three days in cold acclimation conditions may have affected data taken on plants receiving no whole-plant cold stress. Future iterations of such research should aim to record control comparative measurements one day prior to post-stress measurements of plants receiving one cold exposure.

Several differences in cultivar were apparent, with 'FINOLA' expressing greater cold hardiness than 'AutoCBD'. Furthermore, 'FINOLA' also expressed greater ranges of purple pigments in plant tissue than 'AutoCBD', but only in cold-acclimated plants. The higher incidence of purple pigment may be attributed to a relationship between anthocyanin production and cold acclimation, leading to increased cold tolerance [33,34] This relationship may help in explaining why 'FINOLA' experienced lower $\Delta Fv/Fm$ and EL values than 'AutoCBD'. The increased cold tolerance of 'FINOLA' in this trial is partially supported by the findings of Mayer et al. [15], which established that 'FINOLA' was one of nine tested cultivars that displayed the lowest electrolyte leakage when exposed to a 7-day acclimation period at 4 °C. The results of this experiment differed from Mayer's, however, in that plants of both cultivars in this experiment showed greater susceptibility to cold damage in individuals that received cold acclimation treatments, as opposed to an increased tolerance to cold. The inclusion of cold acclimation as a treatment group was intended to increase a plant's potential to tolerate cold exposures, as reported in previous research [35,36]. On the contrary, our data suggest that a 10-day acclimation period at 10 °C did not protect plants from future cold exposures but instead caused greater damage to leaf tissue than a series of 3 h exposures at −0.5 °C.

These results are not surprising, however, when compared with results from the freezing intensity experiment. Plants did not experience a significant effect of cold intensity as measured by any quantification method when exposed to temperatures as low as −4 °C. Cold stress response was only observed in plants in all treatment groups when exposed to −8 °C, which significantly affected every stress quantification method. Similar correlation between increased cold damage and cold-acclimated plants observed in whole-plant cold stress was further observed in this detached-leaf trial, with the exception of electrolyte leakage measurements. Although mean electrolyte leakage differed greatly between cold-acclimated and non-acclimated plants at −8 °C, it was approximately equal between acclimation treatments across all temperature treatments. This observed difference in quantified tissue damage between $\Delta Fv/Fm_{DL}$ and $EL_{DL}$ may be in part attributed to secondary damage experienced by leaves during incubation in distilled water for EC measurements [37].

### 4.2. Cold Temperature Effects on Biomass and Cannabinoid Profile

Overall, cold treatment had very few effects on plant biomass, with the exception of reduced weight in plants that were cold-acclimated. In contrast, cannabinoid profiles were influenced by cold stress. The greatest impact of cold temperatures on cannabinoid content was the decrease in total CBD and THC % when exposed to cold-acclimated conditions.

Total THC concentrations declined more significantly than total CBD concentrations when exposed to 10 °C for 10 days. This pattern of cannabinoid expression was further evidenced when comparing populations of chemotype III plants that exceeded 0.3% total THC in cold-acclimated (0%, n = 0) versus non-acclimated (55%, n = 26) plants.

The mean CBD:THC across all samples was 24.5, in agreement with previously documented mean CBD:THC ranges [3,13]. This trend is supported by extensive literature linking expression of chemotype primarily to genotype [6,38,39]. CBD:THC ratios were also affected by cold stress driven primarily by changes in THC concentrations. CBD:THC ratio was lowest in non-acclimated middle-age plants following three cold stress exposures but was highest following three cold exposures in acclimated plants. Variations in concentrations of CBD and THC are not uncommon in scientific literature (12, 13, 3); however, in many instances, environmental drivers of variation are not substantiated. Cold acclimation had a strong interaction with plant age. Plants exposed to whole-plant cold stresses earliest in their flower development (youngest plant age treatment) expressed lower mean concentrations of all total cannabinoids when compared to older plants but only in cold-acclimated groups. Although literature exists elucidating trends of cannabinoid accumulation, further studies may find value investigating how the variable timing of plant stress during a plant's flower development ultimately affects its accumulation of cannabinoids.

## 5. Conclusions

In conclusion, these results have advanced our understanding of how cold temperatures impair physiological development of hemp and alter intra-chemotype cannabinoid ratios and concentrations found in its flowering structures. Future trials assessing hemp's cold tolerance should evaluate consecutive exposures at lower temperatures and/or greater durations. Furthermore, trials in field settings may advance initial conclusions of this controlled-environment experiment. Work conducted here suggests additional research may be necessary in understanding how timing of other environmental plant stressors—or combinations of stressors—during a plant's flower development ultimately affect its accumulation of cannabinoids. In light of recent advancements in crop insurance technologies, developing an extensive understanding for hemp's capacity to tolerate cold temperatures—particularly early frosts prior to harvest—may help cultivators mitigate the adverse weather effects influencing plant health and cannabinoid profiles [40]. Understanding the impact of a fluctuating global climate on the health and secondary metabolite synthesis of hemp must continue to be prioritized by breeders and cultivators who face the immediate realities of unpredictable climate patterns of the 21st century.

**Supplementary Materials:** The following are available online at https://www.mdpi.com/article/10.3390/horticulturae8060531/s1, Table S1: Whole Plant Raw Data, Table S2: Detached Leaf Raw Data, Table S3: Cannabinoid Raw Data, Figure S1: Histograms of Residuals.

**Author Contributions:** Conceptualization, A.G. and L.B.S.; software, A.G. and H.G.; formal analysis, A.G. and H.G.; investigation, A.G., N.K., H.G. and K.M.; resources, H.G., N.K., W.B.M. and L.B.S.; data curation, A.G. and H.G.; writing—original draft preparation, A.G., H.G. and L.B.S.; writing—review and editing, A.G., H.G., N.K., K.M., W.B.M. and L.B.S.; supervision, H.G., N.K. and L.B.S. All authors have read and agreed to the published version of the manuscript.

**Funding:** This project was partially supported by the New York State Department of Agriculture and Markets through a grant (#132,997) from Empire State Development Corporation.

**Data Availability Statement:** Not applicable.

**Acknowledgments:** We would also like to thank contributions from Cornell staff and students who provided insight, resources, and acquisition of data, including Nick van Eck, Conor Stephen, Alexandria Castaneda, Anthony Barraco III, and Paul Reum. We are grateful to Phylos Bioscience, Inc., for donating seed of 'AutoCBD' and to Endobotanical, LLC, for donating seed of 'FINOLA'.

**Conflicts of Interest:** The authors declare no conflict of interest.

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
