# Peer review of "Effects of Cold Temperature and Acclimation on Cold Tolerance and Cannabinoid Profiles of Cannabis sativa L. (Hemp)"

_horticulturae, doi:10.3390/horticulturae8060531_

Round 1
Reviewer 1 Report
Manuscript ID: horticulturae-1722687
Type of manuscript: Article
Title: Effects of freezing and cold acclimation on cold tolerance and cannabinoid profiles of Cannabis sativa L. (hemp)
Authors: Andrei Galic, Heather Grab, Nicholas S. Kaczmar, Kady Maser, William B. Miller, Lawrence B. Smart
Review of the manuscript
Comments
The manuscript presented for revision is very interesting. This work concerns a important area of science and is of great practical importance. The presented results allow to broaden the knowledge about Cannabis sativa L. capacity to tolerate cold temperatures - particularly early frosts prior to harvest. Conclusions from these studies may help cultivators mitigate the adverse weather effects influencing cannabinoid profiles. The results of these studies may also help in the legislative work is underway on the therapeutic use of cannabis and the reduction of undesirable substances in it.
The obtained results are discussed with the works published in recent years. This work is well organized and scientifically sound.
To sum up, my recommendation is - accept in present form
Reviewer 2 Report
In this paper the effect of cold acclimatization and no acclimatization on subsequent freezing protocols are used to examine if acclimatization can enhance cold tolerance in industrial hemp.
Although this paper appears well presented and planned some problems in experimental design and unexpected results are evident from the authors own comments. For example,
“there were not enough 90 female 'FINOLA' plants to form a treatment group of non-acclimated older plants,” As such it could be pointed out that there would appear to be poor planning of the experiment if insufficient plants were available to carry it out. As a result, there is no data for non-acclimatised ‘oldest’ Finola plants. This means the data set for this condition is incomplete.
“Furthermore, control plants receiv- 413 ing no cold exposures often displayed values of plant stress greater than plants receiving one, two, and even three cold exposures.” And “On the contrary, our data suggests that a 10-day acclimation period at 10° C did not protect plants from future cold exposures but instead caused greater damage to leaf tissue than did a series of 3 hour exposures at -0.5° C.”
The values of ΔFv/FmWP given in the figures are mainly positive which means that according to the formula given (post whole plant cold stress Fv/Fm) – (pre whole plant cold stress Fv/Fm) that the value before plant stress was smaller than after cold stress. This would go against the hypothesis that non-stressed plants exhibit the highest values of this parameter about 0.82 and when stressed this value would be expected to decrease. As such it is unfortunately, difficult to interpretate and evaluate the correctness of the results obtained in this study.
The absence of data for the Finola cultivar “non-acclimatised old plants” and no post-harvest measurements of cannabinoids and biomass suggest that the presentation of data relevant to the ‘Finola’ cultivar should be re-examined or omitted.
The introduction is comprehensive of the topic with respect to the cannabinoids but does not introduce the background information for the methods used to examine plant stress such as chlorophyll fluorescence and electrolyte leakage.
In the text “cold stress” is used. It would be clearer to readers to use periods of freezing in the text as cold stress may not necessarily mean freezing.
These points are covered more specifically below:
Title
“Effects of freezing and cold acclimation on cold tolerance and cannabinoid profiles of Cannabis sativa L. (hemp)”
In the title the term “freezing” is used, whereas in the text “cold stress” is used. It would be clearer to readers to use periods of freezing in the text as cold stress may not necessarily mean freezing.
Keywords:
Consider adding: industrial hemp.
Introduction
The introduction is comprehensive of the topic with respect to the cannabinoids but does not introduce the background information for the methods used to examine plant stress such as chlorophyll fluorescence and electrolyte leakage. Some references and comments are found in the Materials and methods but their interpretation in the introduction is necessary for comprehension of the measurements made. For example, what does Fv/Fm mean? Should its value be high or low if the plant is not stressed. Also, it could be mentioned that some hemp cultivars are more cold tolerant than others.
It briefly lays out the objectives of the study.
Materials and Methods
L 87. According to the text “One week prior to cold stress treatment, a stratified randomization based 87 on height was used to group four biological replicates from each age group into four 88 cold stress groups (n = 16 per group), each receiving a different cold stress treatment. 89 Due to the dioecious nature and low germination of 'FINOLA', there were not enough 90 female 'FINOLA' plants to form a treatment group of non-acclimated older plants, and 91 the acclimated and non-acclimated groups of youngest plants contained 14 'FINOLA' 92 plants each. As result, 96 'AutoCBD' and 76 'FINOLA' plants split between three age 93 groups were divided into four cold stress treatment groups per acclimation treatment. 94”
As such it could be pointed out that there would appear to be poor planning of the experiment if insufficient plants were available to carry it out. Also, this text is not easy to follow to find out how many replicates there were in each treatment.
L 103. “Plants were 102 exposed to -0.5° C for 3 hours in darkness and returned to cold acclimated conditions 103 until harvest.” Is this correct? It appeared that cold stress occurred after acclimatization had finished.
- 111. “Chlorophyll measurements were quantified as the change in Fv/Fm 111 (ΔFv/FmWP) and in SPAD (ΔSPADWP) as measured before and after whole plant cold 112 stress (Table 1). Parameters? Not measurements.
- 119 “Only leaves from experimental units receiving the control 119 whole plant cold stress were used for detached leaf cold stress.” Simplify, its not very clear.
L 122. “Samples were exposed to -2° C on June 4th, 2021, -4° C on June 5th, 2021, and 122 -8° C on June 6th, 2021.” This could be mistaken to mean samples were consecutively exposed to all these temperatures.
L 124. “To quantify chlorophyll damage as a result of cold exposure intensity, initial Fv/Fm 124 and SPAD values were identical to initial values used in quantifying damage resulting 125 from frequency of cold exposure (initial Fv/FmWP and SPADWP). However, in this case, 126 ΔFv/FmDL and ΔSPADDL values were generated using final Fv/Fm and SPAD values col- 127 lected immediately after cold exposure (Table 1).” ‘were identical to” replace with “the same values were used” or similar.
L 131. “Kenneth Post Laboratory: 42.449, -76.468” Explain what these numbers mean?
L 145. “330 umolm-2sec-1 “ Correct to “μmol m-2 sec-1”
- 180. “Samples collected for 180 high pressure liquid chromatography (HPLC)” What did these consist of inflorescences and or leaves?
L 190. “cellulose.” 190 Insert “filter”?
L 196. “quantifi- 196 cation in the range of 1-250 μg/mL-1 and” Correct to μg mL-1
Results
There is no data for non-acclimatised ‘oldest’ Finola plants. This means the data set for this condition is incomplete.
The values of ΔFv/FmWP in the figures are mainly positive which means that according to the formula given (post whole plant cold stress Fv/Fm) – (pre whole plant cold stress Fv/Fm) that the value before plant stress was smaller than after cold stress. This would go against the hypothesis that non-stressed plants exhibit highest values of this parameter about 0.82 and when stressed this value would be expected to decrease. As such it is unfortunately, difficult to interpretate and evaluate the correctness of the results obtained in this study.
Figure 1 Use “Cold acclimatization” in figure annotation instead of “acclimatization”. Also show the period of time and not the start. The black lines with the bent top are confusing and would seem to indicate a period of time (duration).
Also include the non-acclimatized conditions in this Figure. Here “Plant age: P1, P2 and P3” is used. In other Figures “young, middle-aged, and old” are used? This terminology should be consistent throughout. Are the age divisions appropriate as there are only two-week differences between the age groups?
Figure 2 Is the term “acclimated” correct? Maybe “acclimatized”? Y-axis WP should be in smaller case. Why is there no data for non-acclimatized FINOLA old plants? X-axis unit is hour? In the figure the cultivar Finola is written in lower case use FINOLA to be consistent with the rest of the text. whole plant
Legend “Cold stress 235 consisted of no exposure, 0 (control); a single 3 hour -0.5° C exposure, 1; two consecutive 3 hour - 0.5° 236 C exposures separated by 24 hours, 2; and three 3 hour -0.5° C exposures separated by 24 hours, 3.” This explanation should correspond to the values 0, 1, 2, and 3 given on the axis.
“Y axis scale differs between acclima- 239 tion treatments to accurately reflect data.” What does this mean?
Figure 3 Give in full abbreviations used in the legend to the figure. Legend should start with electrolyte leakage to understand what the figure describes. Explain EC and what was measured. y-axis unit.
Figure 4 Detached leaf. The “degree” symbol should be on the unit C not the number (see numbers on x-axis and unit C) see also in the text.
Figure 5 What does SPAD measure?
Figure 6 Why not also for Finola? This is not the MW 0.877. Is this a derivative of the MW?
Figure 2 – 7 Letters 374 indicate differences amongst …..etc not given.
Table 1 Column one heading is not related to a value. These are parameters. Put brackets around the phrases used in the calculations so that the mathematical symbols are more evident. Give abbreviation definitions.
Table 2 Align the asterisks with ns (not superscript) for easier reading
Table 4. “Summary of effects of cultivar, cold acclimation, plant age, whole plant cold stress, their 341 interactions and mean values for post-harvest properties of ‘AutoCBD’ and ‘FINOLA’.” Correct, the text states that “Postharvest data was collected from only ‘AutoCBD’ plants.”
“Cold stress reduced both CBD and THC content but only in the 360 non-acclimated plants exposed to 2 cold stresses (Figure 6, Figure 8). Why was it higher after the third cold stress?
Discussion
L 392 “consecutive exposures 392 to cold temperatures did not consistently result in greater amounts of photoinhibition in 393 any treatment group.” Explain how photoinhibition was determined.
L 396. “suggesting that cold acclimation acted as a 396 plant stress as opposed to a protective, hardening mechanism.” Note that the literature indicates that different cultivars are cultivated according to northern or southern climates
L 413. “Furthermore, control plants receiv- 413 ing no cold exposures often displayed values of plant stress greater than plants receiving 414 one, two, and even three cold exposures.” This appears to be a major problem with this paper as indicated above.
L 420. “Future iterations of such research should aim to record control compara- 420 tive measurements one day prior to post stress measurements of plants receiving one 421 cold exposure.” Indicates that the methodology could be improved.
L 438. “On the contrary, our data suggests that a 10-day acclimation period at 10° C did 438 not protect plants from future cold exposures but instead caused greater damage to leaf 439 tissue than did a series of 3 hour exposures at -0.5° C.” Could this be related to the growth temperatures of the experiment or the time of year that it was carried out? Plants grown at 20 C in the field would probably not experience a ten degree drop in temperature or frost treatment before flowering. Also harvesting occurred in early July.
L 460. “This pattern of cannabinoid expres- 460 sion was further evidenced when comparing populations of chemotype III plants that 461 exceeded 0.3% total THC in cold acclimated (0%, n=0) versus non-acclimated (55%, n = 462 26) plants.” Why was there such a large number of plants exceeding 0.3%?
Conclusions
“In conclusion, these results have allowed us to understand how cold temperatures 480 impair physiological development of hemp and alter intra-chemotype cannabinoid rati- 481 os and concentrations found in its flowering structures.” This is a rather ambitious statement and other factors may have contributed to the observed results.
Reviewer 3 Report
Thank you for the opportunity to review “Effects of freezing and cold acclimation on cold tolerance and cannabinoid profiles of Cannabis sativa L. (hemp)” submitted by Galic et al. In this article, the authors examine how hemp-type cannabis plants respond at the physiological and chemical level to cold and freezing conditions. Their results sometimes differed from the literature in regards to expectations of hardiness from cold acclimation, showing that hemp-type varieties are, somewhat paradoxically, susceptible to prolonged mild cold stress and tolerant of shorter freezing conditions. As presented in this report, these findings are novel and have potential for immediate research/industrial impact.
As a whole, the data that is made available at the time of review generally supports the interpretations of the authors. However, I have concerns with the approach and representation of the data from a statistical perspective. There is not enough information in the statistical analysis section to support that all assumptions required to use ANOVA have been adequately met. Each group has low sample size (4 per group, or less in the case for finola) that appear to contain outliers or are not normally distributed based on the graphs provided. I would like to see the results of Shapiro-Wilk tests and Levene’s test on these groups to justify the use of ANOVA on these data.
I also disagree with the use of box-plots to represent groups with N < 5 as the strength of using box-plots as a device for data representation falls apart with such low sample size (can we really speak of using quartiles and interquartile ranges on n = 4 or less?). I recommend the authors read Krzywinski and Altman (2014; Nature Methods) for more information on the use of box plots and alternative methods of data representation for low sample size groups.
The data collected by the authors will provide valuable insight to an important commercial crop with immense societal value, but I request the authors provide more information to support their current statistical approach, or revise with non-parametric or alternative statistical approaches before I can support its publication.
Below are some minor comments/questions for the authors:
- While I understand the authors choice to represent the cannabinoids from the perspective of their more commonly known decarboxylated forms, I strongly encourage presenting the chemical profiling with the acidic form of all cannabinoids. As it is written the article is incorrectly framed as the plants are producing THC, CBD, and CBG, without clarifying anywhere that plants do not naturally produce these cannabinoids to measurable levels, and that their decarboxylation occurs outside of the normal biological reactions of the plant’s physiology/metabolism.
- Line 199: Can the authors clarify why the presence of an unexpected THC/CBD profile warrants removal from their data set? Cannabis is notoriously ‘leaky’ with respect to cannabinoid profiles and unless the authors believe the variation was the result of technical error, I would argue that a deviating ratio reflects true natural variation and should be included.
.
- Line 408: How was the higher incidence of pest pressure measured, and how was it controlled for through the remainder of the project? Further information regarding the infections and any treatment/quarantining applied should be disclosed.
- Lines 413-422: This seems like a technical limitation that could be described in the methods. The authors do not suggest why waiting an extra four days to take the control’s second measurements would be different from their proposed alternative, and in my opinion does not add to the discussion in a meaningful way.
Round 2
Reviewer 2 Report
The authors have considered thoroughly the comments made in the first review report and have revised the manuscript accordingly.
It is now suitable for publication.
Just a note that the bracketing in Table 1 was suggested to be (phrase)-(phrase) to emphasize the minus sign and not (phrase-phrase) as in the revision made.
Author Response
We thank the reviewer for their support of publication. We have adjusted parentheses to emphasize minus signs as suggested. Thank you.
Reviewer 3 Report
My minor comments/concerns have been satisfactorily addressed. However, my major comments/concerns were not addressed at all. I have copy/pasted these concerns and elaborated on them slightly:
As a whole, the data that is made available at the time of review generally supports the interpretations of the authors. However, I have concerns with the approach and representation of the data from a statistical perspective. There is not enough information in the statistical analysis section to support that all assumptions required to use ANOVA have been adequately met. Each group has low sample size (4 per group, or less in the case for finola) that appear to contain outliers or are not normally distributed based on the graphs provided. I would like to see the results of Shapiro-Wilk tests and Levene’s test on these groups to justify the use of ANOVA on these data.
The assumptions of ANOVA that I am not yet convinced are true, or that the authors have not provided evidence that they have been considered, are that the dependent variables of each combination of groups are normally distributed (can be tested with Shapiro-Wilk) and that each combination of groups have homogeneity of variance (can be tested with Levene's). Please run these tests and include a statement in your statistical methods section that details how you tested the assumptions and whether your data does or does not meet the requirements for ANOVA.
I also disagree with the use of box-plots to represent groups with N < 5 as the strength of using box-plots as a device for data representation falls apart with such low sample size (can we really speak of using quartiles and interquartile ranges on n = 4 or less?). I recommend the authors read Krzywinski and Altman (2014; Nature Methods) for more information on the use of box plots and alternative methods of data representation for low sample size groups.
As previously outlined in Krzywinski and Altman, boxplots require 5 or more samples in each group to be used, and the authors do not have any groups that meet this criteria. The only appropriate way to display your data is by showing the individual data points.
Author Response
We thank the reviewer for their suggestions. Additional context has been added to ln 233 to validate model assumptions for ANOVA.